# Beyond Normalization:
# Rethinking the Partition Function as a Difficulty Scheduler for RLVR

**Dohyung Kim** [1]  **Minbeom Kim** [1]  **Jeonghye Kim** [2]  **Sangmook Lee** [1]  **Sojeong Rhee** [2]  **Kyomin Jung** [1]

## Abstract

Reward-maximizing RL methods have shown to be capable of enhancing the reasoning performance of LLMs, but often lead to reduced generation diversity. Recent works address this issue by adopting GFlowNets, training LLMs to match a target distribution while jointly learning its partition function. In contrast to prior works that treat this partition function solely as a normalizer, we reinterpret it as a per-prompt expected-reward (i.e., online accuracy) signal, leveraging this unused information to improve sample efficiency. Specifically, we first establish a theoretical relationship between the partition function and per-prompt accuracy estimates. Building on this key insight, we propose **Pa**rtition Func**t**ion-Guid**ed RL** (PACED-RL), a post-training framework that leverages accuracy estimates to prioritize informative question prompts during training, and further improves sample efficiency through an accuracy estimate error–prioritized replay. Crucially, both components reuse information already produced during GFlowNet training, effectively amortizing the compute overhead into the existing optimization process. Extensive experiments across diverse benchmarks demonstrate strong performance improvements over GRPO and prior GFlowNet approaches, highlighting PACED-RL as a promising direction for a more sample efficient distribution-matching training for LLMs.

## 1. Introduction

Reinforcement Learning (RL) has emerged as a cornerstone for improving the reasoning performance of Large Language Models (LLMs) (Jaech et al., 2024; Guo et al., 2025; Co-

manici et al., 2025). By enabling models to repeatedly self-explore and maximize the expected reward, such methods have been shown to significantly enhance complex reasoning performances (Shao et al., 2024; Yu et al., 2025b).

Despite these gains, *reward-maximizing* RL methods such as PPO (Schulman et al., 2017) and GRPO (Shao et al., 2024) often lead to an overly sharpened output distribution (Huang et al., 2025a; Li et al., 2025c), leading to limited diversity among generated outputs (Padmakumar & He, 2024; Shypula et al., 2025). In the context of RL post-training for LLM reasoning, however, preserving diversity is essential for a thorough exploration of the search space, enabling the discovery of diverse valid reasoning strategies and the effective application of inference-time techniques (Yue et al., 2025; Li et al., 2025b; Chen et al., 2025).

To alleviate this issue, recent methods such as FlowRL (Zhu et al., 2026) depart from direct reward maximization and instead employ GFlowNets (Bengio et al., 2023), focusing on reward *distribution matching*. Unlike post-training with reward-maximizing RL, where the LLM policy is optimized to maximize the expected reward, FlowRL trains the policy to match a reward-induced target distribution, with a partition function jointly learned as the normalizer of that target reward distribution. At optimum, this partition function corresponds to the sum of the reward-induced distribution mass, accumulated over all possible completions for a given input question prompt. By explicitly modeling the target distribution and pushing the policy toward it, this distributional objective encourages mode coverage, thereby improving upon the diversity limitations of conventional RL methods.

In this work, we revisit the GFlowNet partition function used in LLM post-training. We show that the partition function—previously regarded only as a necessary overhead for target distribution normalization and training stability—naturally encodes per-prompt online accuracy information that can be leveraged for significant improvements in sample efficiency. Specifically, we establish a theoretical connection between the partition function and online accuracies, showing that it can be used directly to estimate the accuracy of a given question prompt under the policy during training. This insight allows us to re-purpose the cost already incurred by GFlowNet training to guide adaptive prompt selection

---

[1]Seoul National University [2]KAIST. Correspondence to: Dohyung Kim <kimdohyung@snu.ac.kr>, Kyomin Jung <kjung@snu.ac.kr>.

*Proceedings of the 43rd International Conference on Machine Learning*, Seoul, South Korea. PMLR 306, 2026. Copyright 2026 by the author(s).

and replay prioritization, focusing on the most informative training samples at each step.

Building on this observation, we propose **Pa**rtition Fun**c**tion-Guid**ed** **RL** (PACED-RL), a novel GFlowNet-based LLM post-training framework that extends the role of the partition function beyond normalization to enable a more sample efficient training, while preserving the inherent diversity benefits of GFlowNets. By making use of the partition function to directly predict accuracy estimates, our method performs difficulty-aware adaptive prompt selection, sampling question prompts that maximize learning efficiency at each step. Furthermore, inspired by prior work on replay mechanisms (Schaul et al., 2015; Shen et al., 2023; Kim et al., 2024), we introduce an accuracy estimation error–prioritized replay strategy that exploits the off-policy tolerance of the GFlowNet objective to further boost sample efficiency. Crucially, both components reuse information *already* produced during standard GFlowNet training, effectively amortizing the cost of adaptive prompt selection and replay prioritization into the existing optimization process.

Extensive experiments across mathematical reasoning and coding benchmarks showcase the effectiveness of PACED-RL. On AIME benchmarks, PACED-RL improves average pass@1 performance by up to 29.1% and 40.0% over GRPO and FlowRL, respectively. Moreover, on the pass@$k$ metric, which we employ as a proxy for diversity and exploration capacity (Li et al., 2025b; Zhu et al., 2025), PACED-RL achieves consistent improvements over baselines, outperforming GRPO and FlowRL by up to 14.2% and 9.1%.

Our main results and key contributions are as follows:

- We theoretically show that the GFlowNet partition function for LLM post-training is linked to online accuracies, enabling it to directly serve as online accuracy estimates.
- We introduce PACED-RL, which leverages these accuracy estimates to perform difficulty-aware adaptive prompt selection and estimation-error–prioritized replay, boosting training efficiency while preserving output diversity.
- We carry out extensive evaluations across code and mathematical reasoning tasks, and show that PACED-RL achieves clear performance gains over recent baselines.

Overall, our results show that PACED-RL offers a principled and practical way to further enhance distribution-matching training of LLMs for reasoning tasks.

## 2. Preliminaries

### 2.1. Reinforcement Learning with Verifiable Rewards

Reinforcement Learning with Verifiable Rewards (RLVR) has emerged as an effective approach for training LLMs on verifiable tasks. Given an output $\mathbf{y}$ generated by an LLM policy $\pi_\theta$ for an input question prompt $\mathbf{x}$ sampled from a dataset $\mathcal{D}$, a deterministic reward function $r(\mathbf{x}, \mathbf{y}) \in \{0, 1\}$ assigns a binary score to $\mathbf{y}$ by assessing its correctness. RLVR optimizes the following KL-regularized objective:

$$\max_\theta \mathop{\mathbb{E}}_{\substack{\mathbf{x} \sim \mathcal{D} \\ \mathbf{y} \sim \pi_\theta(\cdot | \mathbf{x})}} [r(\mathbf{x}, \mathbf{y})] - \beta D_{\mathrm{KL}}(\pi_\theta(\cdot | \mathbf{x}) \| \pi_{\mathrm{ref}}(\cdot | \mathbf{x})) \quad (1)$$

, where $\pi_{\mathrm{ref}}$ denotes the untrained base reference model and $\beta$ denotes the KL divergence regularization coefficient.

In particular, the optimal policy for Eq. 1 admits a closed-form solution, expressed as:

$$\pi^*(\mathbf{y} | \mathbf{x}) = \frac{\pi_{\mathrm{ref}}(\mathbf{y} | \mathbf{x}) \exp(\beta^{-1} r(\mathbf{x}, \mathbf{y}))}{Z(\mathbf{x})}. \quad (2)$$

Here, $Z(\mathbf{x})$ denotes the intractable partition function that normalizes the distribution by summing over all possible outputs $\mathbf{y}$ for a given question prompt $\mathbf{x}$:

$$Z(\mathbf{x}) = \sum_{\mathbf{y}} \pi_{\mathrm{ref}}(\mathbf{y} | \mathbf{x}) \exp(\beta^{-1} r(\mathbf{x}, \mathbf{y})). \quad (3)$$

### 2.2. GFlowNets for LLM Post-Training

GFlowNets train a policy $\pi_\theta$ to sample diverse discrete, compositional objects in proportion to an unnormalized target reward *distribution* $R(\mathbf{x}, \mathbf{y})$. In the RLVR setting, this objective biases the policy to sample high-reward outputs $\mathbf{y}$ (correct solutions) rather than low-reward outputs (incorrect solutions) for a given input question prompt $\mathbf{x}$.

To train an LLM policy $\pi_\theta$ to sample from the optimal policy in Eq. 2 using GFlowNet objectives, previous works (Lee et al., 2025; Zhu et al., 2026) configure the unnormalized reward distribution as $R(\mathbf{x}, \mathbf{y}) = \pi_{\mathrm{ref}}(\mathbf{y} | \mathbf{x}) \exp(\beta^{-1} r(\mathbf{x}, \mathbf{y}))$, and train a learnable $Z_\phi(\mathbf{x})$, which approximates the intractable partition function $Z(\mathbf{x})$ in Eq. 3. By explicitly modeling the target distribution and its normalization through a learnable partition function, GFlowNets reduce policy learning to matching this target distribution.

Plugging this $R(\mathbf{x}, \mathbf{y})$ and $Z_\phi(\mathbf{x})$ into the Trajectory Balance (TB) objective (Malkin et al., 2022) for GFlowNet training leads to the loss function defined as follows:

$$\mathcal{L}_{\mathrm{TB}}(\mathbf{x}, \mathbf{y}; \theta, \phi) = \left[ \log \left( \frac{Z_\phi(\mathbf{x}) \pi_\theta(\mathbf{y} | \mathbf{x})}{\pi_{\mathrm{ref}}(\mathbf{y} | \mathbf{x}) \exp(\beta^{-1} r(\mathbf{x}, \mathbf{y}))} \right) \right]^2. \quad (4)$$

Minimizing this TB loss is equivalent, in terms of expected gradients, to minimizing the KL divergence between $\pi_\theta$ and the optimal policy in Eq. 2, with the intractable partition function approximated by $Z_\phi(\mathbf{x})$ (Zhu et al., 2026):

$$\min_\theta \mathcal{L}_{\mathrm{TB}}(\mathbf{x}, \mathbf{y}; \theta, \phi) \iff$$

$$\min_\theta D_{\mathrm{KL}} \left( \pi_\theta(\mathbf{y} | \mathbf{x}) \,\middle\|\, \frac{\pi_{\mathrm{ref}}(\mathbf{y} | \mathbf{x}) \exp(\beta^{-1} r(\mathbf{x}, \mathbf{y}))}{Z_\phi(\mathbf{x})} \right). \quad (5)$$

Thus, $\mathcal{L}_{\text{TB}}$ drives the LLM policy $\pi_\theta$ to sample from the optimal policy, while simultaneously training $Z_\phi(\mathbf{x})$ to normalize the unnormalized target reward distribution $\pi_{\text{ref}}(\mathbf{y} \mid \mathbf{x}) \exp\big(\beta^{-1}r(\mathbf{x}, \mathbf{y})\big)$.

## 3. Related Works

### 3.1. GFlowNets for LLM Post-Training

In the context of LLM post-training, GFlowNets have been adapted for domains such as preference alignment, red-teaming, and reasoning (Kwon et al., 2024; Lee et al., 2025; Yu et al., 2025a; Zhu et al., 2026). Bartoldson et al. (2025) further proposes an asynchronous, distributed RL pipeline for LLMs that leverages GFlowNets for better throughput.

Despite their success, existing GFlowNet-based methods for LLM post-training primarily treat the partition function as a necessary normalization variable required to define the target distribution. While several works study improved optimization techniques for GFlowNets, such as replacing the learned $Z_\phi(\mathbf{x})$ with a batch-estimate (Zhang et al., 2023; Bartoldson et al., 2025), the information encoded in $Z_\phi(\mathbf{x})$ has not been exploited beyond normalization.

Most closely related to our work is FlowRL (Zhu et al., 2026). FlowRL adapts GFlowNets to the synchronous RL setting, where training alternates between the rollout generation phase and the optimization phase, with added stability techniques suited for RLVR. Our work extends prior works in GFlowNets for LLM post-training by reformulating $Z_\phi(\mathbf{x})$ for online accuracy estimation, enabling adaptive prompt selection and replay strategies that enhance sample efficiency without incurring additional cost.

### 3.2. Adaptive Prompt Selection for RLVR

Recent works show that training on question prompts of intermediate difficulty—those achieving approximately 0.5 accuracy with respect to the policy—improves sample efficiency in the RLVR setting (Bae et al., 2026; Foster et al., 2026). Building on this observation, adaptive prompt selection methods that selectively sample such prompts for training have emerged as a promising direction for improved sample efficiency. Crucially, to effectively enable such methods, obtaining reliable online accuracy estimates is needed.

Recent works such as Yu et al. (2025b); Foster et al. (2026); Zhang et al. (2025) over-sample a pool of question prompts larger than the training batch at each step, and filter out less informative prompts based on the observed accuracies after the rollout generation phase. Though effective, such approaches lead to a significantly increased number of rollout generation, substantially increasing computational overhead. An alternative line of work maintains per-question accuracy histories, and estimates online accuracies via Bayesian pos-

terior estimation or probabilistic filtering (Zheng et al., 2025; Qu et al., 2026; Zeng et al., 2026), enabling lightweight prompt selection. However, on large datasets, these estimates can suffer from off-policy bias: over the course of an epoch, the policy may improve substantially, rendering the accuracy estimates stale when revisited.

Distinct from previous approaches, our work reuses the partition function inherent to GFlowNets for online accuracy estimation, eliminating the need for employing such auxiliary mechanisms to guide adaptive prompt selection.

### 3.3. Prioritized Experience Replay

Replay is widely adopted in RL training, owing to its efficacy in enhancing sample efficiency. To further enhance replay effectiveness, Schaul et al. (2015) propose Prioritized Experience Replay (PER) for Q-learning, which prioritizes samples with large temporal-difference errors, allowing the policy to focus learning on more informative data. Building on this idea, numerous PER variants (Horgan et al., 2018; Hessel et al., 2018; Sujit et al., 2023) have been developed to improve training stability and data efficiency.

Replay mechanisms have also been extensively adopted in the GFlowNets literature (Shen et al., 2023; Kim et al., 2024; Bartoldson et al., 2025), where the training objective naturally accommodates off-policy data, and more recently, have been explored in the RLVR setting (Wang et al., 2025b; Li et al., 2025a). A recurring insight across both domains is that prioritizing high-reward and more recent samples improves the effectiveness of the replay. Extending upon previous works, our work tailors prioritization to improve partition function learning and sample efficiency by leveraging its connection to accuracy estimates and prioritizing samples with large accuracy estimation errors.

## 4. PACED-RL

We begin by revisiting the role of the GFlowNet partition function for LLM post-training. While the learnable partition function $Z_\phi$ is typically introduced as a necessary normalization term, we reveal that it in fact encodes meaningful information about per-question online accuracies. Specifically, we first show that for a question $\mathbf{x}$, $Z_\phi$ provides an estimate of the accuracy $p_{\text{old}}(\mathbf{x})$. Here, $p_{\text{old}}(\mathbf{x})$ denotes the accuracy for $\mathbf{x}$ under the pre-update policy $\pi_{\text{old}}$, the policy in which fresh rollouts are sampled from at each training step. We then show how these accuracy estimates can be leveraged to improve sample efficiency by focusing on the most informative samples throughout the training process.

### 4.1. Partition Function and Online Accuracy Estimates

Under a Trajectory Balance loss modified from Eq. 4, we show that the optimal partition function admits a direct rela-

tionship to per-question online accuracy estimates. Specifically, we replace $\pi_{\text{ref}}$ with $\pi_{\text{old}}$:

$$\mathcal{L}_{\text{ours}}(\mathbf{x}, \mathbf{y}; \theta, \phi) = \left[ \log \left( \frac{Z_\phi(\mathbf{x})\, \pi_\theta(\mathbf{y} \mid \mathbf{x})}{\pi_{\text{old}}(\mathbf{y} \mid \mathbf{x})\, \exp\!\big(\beta^{-1} r(\mathbf{x}, \mathbf{y})\big)} \right) \right]^2.$$
(6)

Replacing the reference policy $\pi_{\text{ref}}$ with $\pi_{\text{old}}$ yields an interpretation as training to sample from the optimal policy of a KL-regularized objective (Eq. 1), with the KL divergence regularization anchored at $\pi_{\text{old}}$ rather than $\pi_{\text{ref}}$.

**Proposition 4.1.** *Given* $Z^*(\mathbf{x})$, *the optimal partition function, we can express* $p_{\text{old}}(\mathbf{x})$ *as follows:*

$$p_{\text{old}}(\mathbf{x}) = \beta \log Z^*(\mathbf{x}) - \beta D_{\text{KL}}\big(\pi_{\text{old}}(\cdot \mid \mathbf{x}) \big\| \pi_\theta(\cdot \mid \mathbf{x})\big). \quad (7)$$

The detailed derivations are provided in Appendix A. Proposition 4.1 reveals that the partition function, previously used solely for the normalization of the target distribution, can be directly interpreted as an online accuracy signal. Importantly, this information is already produced as part of the GFlowNet training, incurring no extra computational cost.

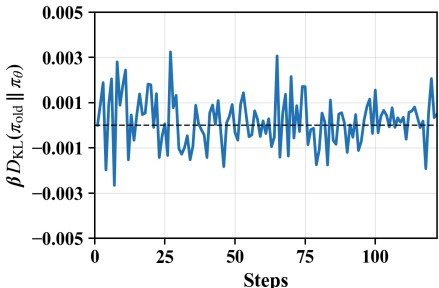

*Figure 1.* Training dynamics of mean of estimated values of $\beta\, D_{\text{KL}}\big(\pi_{\text{old}}(\cdot \mid \mathbf{x}) \big\| \pi_\theta(\cdot \mid \mathbf{x})\big)$ of Qwen2.5-Math-1.5B trained on the DeepScaleR dataset, using the TB loss defined in Eq. 6

### 4.2. Obtaining Practical Online Accuracy Estimates

Ideally, the accuracy estimate for $\mathbf{x}$ should be obtainable as a function of $\mathbf{x}$ without requiring additional computation. While Eq. 7 provides theoretical insight, it does not immediately yield a practical online accuracy estimate from $Z_\phi$, as KL divergence between policies is typically estimated using expensive Monte Carlo rollouts (Schulman et al., 2015). Below, we show that the KL term in Eq. 7 remains uniformly small during training and can therefore be safely omitted, yielding a simple accuracy estimator dependent only on $Z_\phi$.

In practice, standard training practices—such as small learning rates and gradient norm clipping—generally ensure that parameter updates remain small. As KL divergence admits a second-order approximation that scales quadratically with the parameter update (Achiam et al., 2017; Martens, 2020), small parameter updates typically lead to a controlled KL divergence between successive policies.

As shown in Fig. 1, the empirically estimated mean values of $\beta\, D_{\text{KL}}\big(\pi_{\text{old}}(\cdot \mid \mathbf{x}) \big\| \pi_\theta(\cdot \mid \mathbf{x})\big)$ remains uniformly small over the course of training, with the maximum absolute value remaining below $4 \times 10^{-3}$. Since the range of online accuracies $p_{\text{old}}(\mathbf{x})$ is $[0, 1]$ due to the 0/1 binary reward setting, the error that would be introduced by discarding the KL divergence term is small at the scale relevant for difficulty-aware prompt selection.

Motivated by this observation, we obtain a practical biased estimator by discarding the KL divergence term in Eq. 7:

$$p_{\text{old}}(\mathbf{x}) \approx \beta \log Z^*(\mathbf{x}) \quad (8)$$

and train $Z_\phi(\mathbf{x})$, which approximates the optimal $Z^*(\mathbf{x})$. While this approximation introduces an empirically stable and controlled bias into the accuracy estimates, it enables per-question online accuracy estimation without auxiliary estimators or additional rollouts. Consequently, the cost of accuracy estimation is effectively amortized into the existing GFlowNet training process.

### 4.3. Adaptive Prompt Selection Using Online Accuracy Estimates

Having obtained online accuracy estimates directly from $Z_\phi$ *without* incurring additional cost, we can naturally integrate adaptive prompt selection into the training pipeline.

Prior work has shown that question prompts of intermediate difficulty provide the most sample-efficient learning signal (Bae et al., 2026; Foster et al., 2026). Leveraging the online accuracy estimates from $Z_\phi$, we therefore bias training toward question prompts whose predicted accuracies lie in the vicinity of $0.5$, ensuring that the policy consistently focuses on the most informative prompts during training.

Specifically, at training step $t$, we begin by utilizing Eq. 8 to obtain $\{\hat{p}_{\text{old}}(\mathbf{x}_i)\}_{i=1}^{|\mathcal{D}|}$, the estimates of the online accuracies for question prompts in the dataset. Similar to previous works (Yue et al., 2025; Foster et al., 2026), we then greedily select the top-$m$ questions whose estimated accuracies are closest to the target accuracy $\tau$, where $m$ denotes the training batch size. The selected $m$ questions are then used for the $t$-th training step. While we fix $\tau$ to $0.5$ for optimal sample efficiency in the main experiments, we report results for other configurations of $\tau$ in Sec. 5.3.

Following the FlowRL pipeline, for each of the $m$ selected questions, we then generate $N$ rollouts and update both the policy $\pi_\theta$ and the partition function $Z_\phi$ by minimizing Eq. 6.

### 4.4. Accuracy Estimation Error-Prioritized Replay

To further improve sample efficiency and the calibration of $Z_\phi$, we introduce an accuracy estimation error–prioritized replay strategy. Unlike prior replay prioritization approaches (Schaul et al., 2015; Shen et al., 2023), our method

exploits the connection between $Z_\phi$ and online accuracy estimates by focusing on samples with large estimation errors. Intuitively, these samples provide the most informative learning signal for $Z_\phi$; this leads to better-calibrated accuracy estimates and a more accurate normalization of the target reward distribution, thereby improving training.

Specifically, we maintain a replay buffer that stores prompt–output pairs $\{\mathbf{x}, \mathbf{y}\}$. Following prior findings that replay is most effective with high-reward and recent samples (Bartoldson et al., 2025; Li et al., 2025a), we restrict the buffer to outputs satisfying $r(\mathbf{x}, \mathbf{y}) = 1$. Among retained pairs, we prioritize pairs associated with question prompts $\mathbf{x}$ with the highest accuracy estimation error:

$$\text{priority}(\mathbf{x}) = \left| \frac{N_{\text{correct}}}{N} - \hat{p}_{\text{old}}(\mathbf{x}) \right|, \qquad (9)$$

, where $\frac{N_{\text{correct}}}{N}$ denotes the observed accuracy for question $\mathbf{x}$ computed from $N$ rollouts generated by $\pi_{\text{old}}$ during the rollout generation process of the training pipeline.

We utilize a fixed-capacity replay buffer of size $B_{\max}$ into which $B_{\text{add}}$ new samples are added at every training step. This design bounds the age of stored trajectories and thereby controls the degree of off-policyness of the buffer (Fedus et al., 2020). We then augment the training batch with the contents stored in the replay buffer at each training step.

A complete description of our full PACED-RL algorithm, consisting of adaptive prompt selection and accuracy estimate error-prioritized replay, is shown in Algorithm 1.

# 5. Experiments

## 5.1. Experimental Setup

**Datasets & Models.** We conduct experiments in the code generation domain and the mathematical reasoning domain. For code generation, we train on the DeepCoder dataset (Luo et al., 2025a; Zhu et al., 2026) using the DeepSeek-R1-Distill-Qwen-1.5B model (Guo et al., 2025). For mathematical reasoning, we use the DeepScaleR dataset (Luo et al., 2025b) and adopt Qwen2.5-Math-1.5B and Qwen2.5-Math-7B (Yang et al., 2024) as base models. Across all runs, PACED-RL is trained with $\beta = 0.05$ and replay buffer capacity $B_{\max}$ of 128, with $B_{\text{add}} = 64$ prompt-output pairs added to the buffer at each step. Similar to prior works (Lee et al., 2025; Zhu et al., 2026), we use a 3-layer MLP stacked on top of frozen, pre-computed reference model embeddings of question prompts $\mathbf{x}$ to parameterize $Z_\phi$.

**Training & Evaluation.** We adopt the widely used verl library (Sheng et al., 2025) for training. Across all runs, we fix the learning rate to 1e-6 and generate 8 rollouts per question prompt, i.e. $N = 8$. Due to compute resource constraints, we set the maximum generation length to 3072 tokens for

all runs. For code generation, we evaluate on HumanEval+ (Chen, 2021) and LiveCodeBench (Jain et al., 2025), and report the pass@1 performance averaged over 8 attempts throughout training. For evaluation on the mathematical reasoning domain, we use the MATH500 (Hendrycks et al., 2021), MinervaMath (Lewkowycz et al., 2022), Olympiad-Bench (He et al., 2024), and AIME24/25 benchmarks. Similar to prior works (Wang et al., 2025b; Zheng et al., 2025), we evaluate models every 10 steps and report the pass@1 and pass@$k$ performance of the checkpoint with the best average performance across benchmarks. Hyperparameters and other implementational details are in Appendix B.

**Baselines.** To comprehensively evaluate PACED-RL, we compare against representative baselines.
**(1) No Adaptive Prompt Selection:** We include **GRPO** and **FlowRL** as standard baselines that sample prompts uniformly from the training dataset. Neither method performs adaptive prompt selection; **GRPO** represents the reward-maximization approach, while **FlowRL** represents the distribution-matching algorithm based on GFlowNets.
**(2) With Adaptive Prompt Selection:** We include Dynamic Sampling (**DS**) (Yu et al., 2025b) and **LILO** (Foster et al., 2026) as baselines that perform adaptive prompt selection by over-sampling $M > m$ question prompts from the dataset at each training step. After generating rollouts for each of the $M$ over-sampled prompts, DS discards question prompts with observed accuracies of 0 or 1, whereas LILO selects the top-$m$ question prompts whose accuracies are closest to 0.5. We also compare PACED-RL against **MoPPS** (Qu et al., 2026), which estimate prompt accuracies without additional over-sampling. MoPPS maintains per-question prompt accuracy histories and applies Bayesian posterior estimation, selecting the top-$m$ question prompts whose estimated accuracies are closest to 0.5 at each training step.

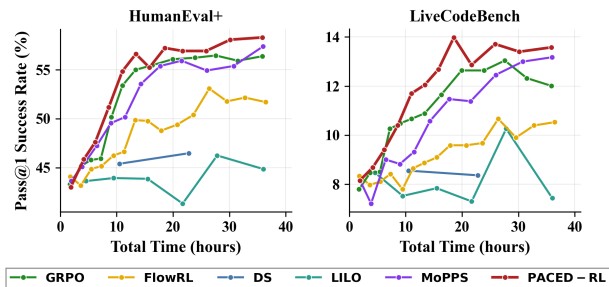

*Figure 2.* Pass@1(%) dynamics of DeepSeek-R1-Distill-Qwen-1.5B plotted with respect to wall-clock time. Pass@1 values obtained via averaging over 8 independent attempts.

## 5.2. Main Results

**Code Generation.** As shown in Fig. 2, PACED-RL consistently achieves the highest pass@1 performance through-

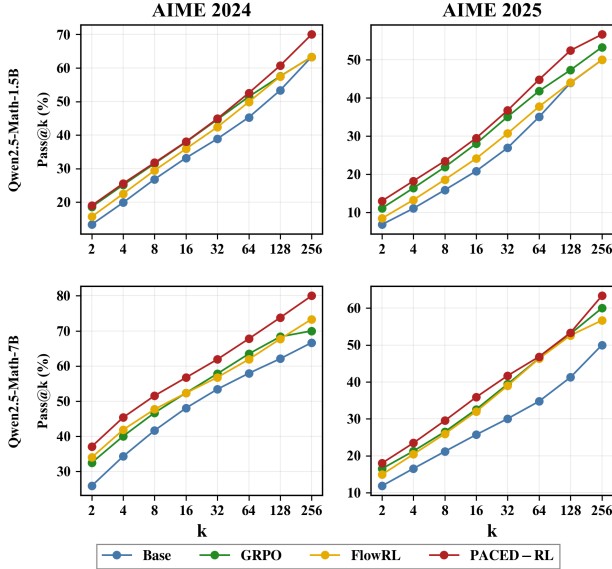

*Figure 3.* Pass@k(%) curves of Qwen2.5-Math-7B trained on GRPO, FlowRL, and PACED-RL and evaluated on AIME 24 and AIME24 for $k \in \{2, 4, 8, 16, 32, 64, 128, 256\}$. Across all evaluated values of $k$, PACED-RL achieves the highest performance.

out training compared to all baselines, illustrating the benefits of improved sample efficiency via leveraging accuracy estimates from the partition function. Notably, PACED-RL achieves significantly faster training than GRPO and FlowRL. On HumanEval+, PACED-RL reaches the best performance achieved by GRPO and FlowRL using only $0.49\times$ and $0.42\times$ of their respective training time. Similarly, on LiveCodeBench, PACED-RL achieves the best performance of GRPO and FlowRL in $0.67\times$ and $0.42\times$ of the time.

Our results also indicate that for code generation tasks, adaptive prompt selection methods that utilize over-sampling question prompts, namely DS and LILO, are particularly ineffective. Unlike the mathematical reasoning domain, where fast rule-based verification is possible, the code generation domain requires slow execution-based verification (Wang et al., 2025a). The verification of increased number of code generations to estimate accuracies required in the DS and the LILO settings leads to prohibitively slow training — leading to sub-optimal results with respect to training latencies.

**Mathematical Reasoning.** Results in Table 1 underscore the effectiveness of PACED-RL in the mathematical reasoning domain. Crucially, PACED-RL consistently outperforms GRPO and FlowRL by a wide margin. On the Qwen2.5-Math-7B model, PACED-RL improves over the performance of GRPO and FlowRL by 29.1% and 40.0% respectively on the AIME 24 benchmark, providing strong evidence of the effectiveness of our method. PACED-RL also consistently matches or exceeds the performance of

adaptive prompt selection baselines, despite not incorporating explicit auxiliary mechanisms for obtaining online accuracy estimates during training. This contrast is particularly notable given that PACED-RL does not incur significant compute overhead from rollout generation, as with DS and LILO. The improvement is most pronounced on the hardest AIME benchmarks: for instance, on AIME 25 with Qwen2.5-Math-7B, PACED-RL attains 13.1 compared to 11.7 for LILO, corresponding to an 11.9% relative gain.

**Diversity.** As a proxy for diversity, we assess the pass@$k$ performance of PACED-RL. Achieving high pass@$k$ requires the model to cover a broader space of plausible solution trajectories rather than concentrating probability mass on a single mode. Consequently, improvements in pass@$k$—particularly at larger $k$—are widely interpreted as evidence of increased solution diversity and exploration (Li et al., 2025b; Chen et al., 2025; Zhu et al., 2025).

We compare the pass@$k$ performance of PACED-RL compared to GRPO, FlowRL, and the untrained base models Qwen2.5-Math-1.5B and 7B. We evaluate on the AIME24/25 benchmarks, and generate using temperature set to 0.6 and a top-$p$ value of 0.95, following Yue et al. (2025). Fig. 3 shows that PACED-RL consistently outperforms baselines across values of $k$, with relative improvements over GRPO and FlowRL by up to 14.2% and 9.1% respectively. These results demonstrate the diversity-preserving nature and the superior exploration capability of PACED-RL.

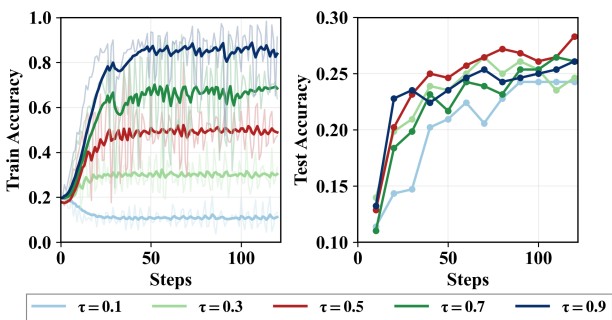

*Figure 4.* Train and test accuracies of Qwen2.5-Math-1.5B trained on the DeepScaleR dataset and evaluated on the MinervaMath benchmark, for target accuracy values $\tau \in \{0.1, 0.3, 0.5, 0.7, 0.9\}$.

## 5.3. Analyses

**Controlling Train Accuracy with the Partition Function.** We vary the target accuracy value $\tau$ to examine whether it enables control over the induced training accuracy and to assess its impact on the training dynamics. Fig. 4 shows that our method selectively retains training questions whose estimated accuracies are close to $\tau$, thereby providing control over the training accuracy. Furthermore, our results reaffirm

*Table 1.* Pass@1(%) performance across the five mathematical reasoning benchmarks. We report the Avg@32 performance for AIME 24 and AIME 25 benchmarks due to their small size. **Avg.** denotes the average of the performance on each of the benchmarks. **Avg. Rollouts** denotes the average number of rollouts generated per training step relative to GRPO, expressed as a multiplicative factor. Best results are in **bold**. Second best results are underlined.

| Model | Method | MATH-500 | OlympiadBench | Minerva | AIME 24 Avg@32 | AIME 25 Avg@32 | Avg.↑ | Avg. Rollouts↓ |
|---|---|---|---|---|---|---|---|---|
| | GRPO | 70.6 | 34.2 | 24.2 | 10.3 | 6.3 | 29.1 | **1.0×** |
| | FlowRL | 67.6 | 32.4 | 20.2 | 9.5 | 5.9 | 27.1 | **1.0×** |
| Qwen2.5-Math-1.5B | DS | 72.8 | 36.8 | 28.3 | 11.8 | 6.5 | 31.2 | 2.0× |
| | LILO | **73.2** | 36.2 | 27.9 | 12.0 | 6.9 | 31.2 | 4.0× |
| | MoPPS | 71.6 | 36.6 | 26.1 | 11.4 | 6.6 | 30.4 | **1.0×** |
| | PACED-RL | **73.2** | **37.9** | **29.0** | **13.3** | **7.3** | **32.1** | **1.0×** |
| | GRPO | 77.6 | 39.7 | 34.9 | 23.9 | 10.3 | 37.2 | **1.0×** |
| | FlowRL | 76.4 | 39.6 | 32.3 | 24.9 | 9.4 | 36.5 | **1.0×** |
| Qwen2.5-Math-7B | DS | **80.2** | 44.0 | 34.5 | 26.6 | 12.0 | 39.4 | 2.5× |
| | LILO | 78.4 | **45.2** | **37.5** | 28.0 | 11.7 | **40.1** | 4.0× |
| | MoPPS | **80.2** | 42.2 | 34.1 | 24.0 | 12.9 | 38.6 | **1.0×** |
| | PACED-RL | 80.0 | 44.6 | 34.1 | **28.7** | **13.1** | **40.1** | **1.0×** |

results from previous works (Foster et al., 2026; Gao et al., 2023) that maintaining questions with accuracies closest to 0.5 is optimal for sample-efficiency, achieving the highest test accuracy throughout training. Setting $\tau = 0.9$ yields the fastest *initial* gains in test accuracy but quickly plateaus, ultimately resulting in sub-optimal performance. Moreover, the run with $\tau = 0.1$ produces the weakest results overall, with test accuracy consistently trailing all other settings throughout training. Taken together, these results indicate that by restricting training to an effective level of difficulty, adaptive prompt selection makes policy learning robust to dataset difficulty, enabling efficient and stable improvement on datasets of any difficulty via a sustained focus on the most informative question prompts.

**On Discarding the KL Divergence Term.** We further empirically show that omitting the KL divergence term from the full expression in Eq. 7 to obtain the practical accuracy estimator in Eq. 8 does not significantly alter the sampling dynamics. Every 5 training steps, we randomly sample 512 prompts from $\mathcal{D}$ and compare the prompts selected by the two expressions. Let $\mathcal{S}_{\text{omit}}$ and $\mathcal{S}_{\text{full}}$ denote the top-$m$ prompts whose estimated accuracies are closest to $0.5$ under Eq. 8 and Eq. 7, respectively. We report the selection overlap $\frac{|\mathcal{S}_{\text{omit}} \cap \mathcal{S}_{\text{full}}|}{k}$, which ranges from 0 to 1, with 1 indicating identical selections. For Eq. 7, the KL divergence term is obtained through Monte Carlo estimation using 8 rollouts, and as in the main experiments, we set $k = 128$.

Results in Figure. 5 show that $\mathcal{S}_{\text{omit}}$ and $\mathcal{S}_{\text{full}}$ exhibit near-perfect overlap throughout training, indicating that omitting the KL divergence term has negligible effect on the sampling dynamics. Furthermore, estimation of the KL divergence term introduces substantial overhead, increasing training-step latency by 70% for the 1.5B model and 87% for the 7B

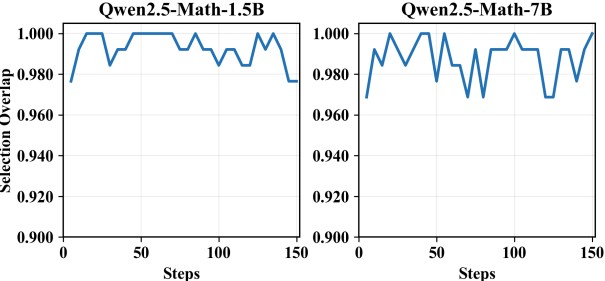

*Figure 5.* Selection overlap between $\mathcal{S}_{\text{omit}}$ and $\mathcal{S}_{\text{full}}$ over the course of training. Results obtained using Qwen2.5-Math-1.5B and Qwen2.5-Math-7B models trained on the DeepScaleR dataset.

model, despite being computed over only 512 prompts rather than the full dataset composed of 40k question prompts. These results illustrate that omitting the KL divergence term is a practical approximation for efficiently exploiting the partition function's accuracy signal, in the domain post-training LLMs for code and mathematial reasoning tasks.

**Computational Overhead.** Owing to the small size of the 3-layer MLP on top of pre-computed reference model embeddings of question prompts $\mathbf{x}$ used to parameterize $Z_\phi$, the computation of $Z_\phi$ for all question prompts in the dataset can be efficiently performed using large-batch inference. To illustrate the negligible computational time overhead, we report in Table 2 the average time required to compute $Z_\phi(\mathbf{x})$, and thus the accuracy estimates $\{\hat{p}_{\text{old}}(\mathbf{x}_i)\}_{i=1}^{|\mathcal{D}|}$, alongside the average total runtime per training step. Across all models and training domains, the average time taken to obtain $Z_\phi(\mathbf{x})$ values is several orders of magnitude smaller than the average time taken per training step.

By exploiting these low-cost accuracy estimates obtainable

*Table 2.* Comparison between the average time taken for the computation of $Z_\phi(\mathbf{x})$ for all question prompts in the dataset and the average total time taken per step. DS-Distill-1.5B denotes the DeepSeek-R1-Distill-Qwen-1.5B model.

| Model | $Z_\phi(\mathbf{x})$ Computation (s) | Total (s) |
|---|---|---|
| Qwen2.5-Math-1.5B | 0.035 | 308 |
| Qwen2.5-Math-7B | 0.110 | 370 |
| DS-Distill-1.5B | 0.020 | 1086 |

with minimal additional latency, PACED-RL substantially improves sample efficiency, translating into notable performance gains.

**Progression of Question Difficulty During Training.**
Figure 6 shows the evolution of the mean difficulty of sampled question prompts over the course of training. For the MATH dataset, we utilize the human-annotated difficulty labels, which range from level 1, indicating the simplest questions, to level 5, indicating the most complex questions. For DeepScaleR, which does not have human-annotated difficulty labels, we use the difficulty labels from Shi et al. (2025), defined as 1 - pass@1 accuracy computed over all questions using the Qwen2.5-Math-7B model. Without adaptive prompt selection, the mean difficulty remains constant throughout training. However, with adaptive prompt selection guided by partition function–guided accuracy estimates, the mean difficulties progressively increase, shifting toward harder questions as the model improves. At latter stages of training, PACED-RL selectively trains on the hardest subsets of the dataset. This trend is expected, as a question regarded as having intermediate difficulty in earlier parts of training is likely to become easier for the policy as training progresses.

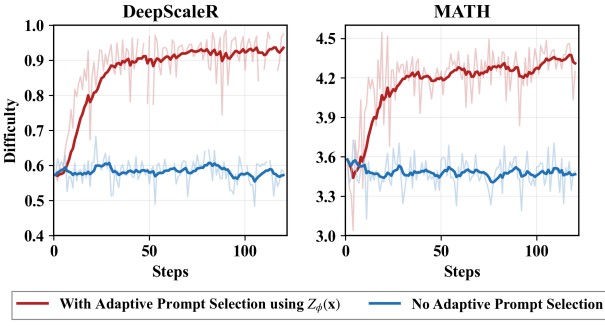

*Figure 6.* Progression of the average difficulty of question prompts sampled in each training batch under the Qwen2.5-Math-7B model.

Such progression of question difficulty naturally prevents PACED-RL from collapsing to a small subset of the dataset during training, despite it greedily selecting the most informative prompts at each training step. We additionally report *relative coverage*, defined as the ratio of unique prompts

*Table 3.* Relative coverage and total number of distinct prompts observed during 150 training steps for the Qwen2.5-Math-1.5B and 7B models trained on the DeepScaleR dataset.

| Metric | Qwen2.5-Math-1.5B | Qwen2.5-Math-7B |
|---|---|---|
| Relative Coverage | 55.7% | 62.3% |
| Distinct Prompts | 10695 | 11962 |

observed by PACED-RL relative to the no-adaptive-prompt-selection variant, together with the total number of distinct prompts encountered during training. As shown in Table. 3, PACED-RL trains on a sizeable subset of the prompts seen by the no-adaptive-prompt-selection variant, filtering out less informative prompts to avoid unnecessary trajectory generation and optimization. Moreover, the number of distinct prompts remains large ($> 10$k), indicating that PACED-RL does not collapse to a narrow subset of the dataset.

**Ablation on the Sampling Mechanism.** PACED-RL greedily selects the top-$m$ prompts whose estimated accuracies are the closest to $0.5$ to maximize sample efficiency. To assess the sensitivity of the training dynamics of PACED-RL to this deterministic prompt selection rule, we additionally evaluate a soft-sampling variant. Specifically, we replace the greedy top-$m$ selection with sampling from a softmax distribution whose logits are defined as $\hat{p}_{\text{old}}(\mathbf{x})(1 - \hat{p}_{\text{old}}(\mathbf{x}))$, with temperature $T$. Since the logits are maximized at $\hat{p}_{\text{old}}(\mathbf{x}) = 0.5$, this configuration naturally assigns higher sampling probability to prompts with estimated accuracies closer to $0.5$, while allowing prompts in a broader neighborhood around $0.5$ to be sampled.

*Table 4.* MATH-500 Avg@8 performance using different prompt sampling strategies. Results obtained with the Qwen2.5-Math-1.5B model trained on the DeepScaleR dataset.

| Temperature | $T = 1.0$ | $T = 0.7$ | $T = 0.4$ | Greedy |
|---|---|---|---|---|
| MATH-500 Avg@8 | 72.2 | 71.4 | 71.8 | 73.2 |

Table. 4 reports Avg@8 performance on MATH500 for $T \in \{0.4, 0.7, 1.0\}$. We observe that soft sampling consistently underperforms the default greedy top-$m$ selection. One possible explanation is that soft sampling assigns non-negligible probability to less informative prompts, leading to such prompts being inserted into the training batch. The quality of the training signal may thus be reduced, and could lead to weaker downstream performances.

**Ablation on the Replay Mechanism.** To isolate the contribution of our accuracy estimate error–prioritized replay mechanism, we conduct an ablation study on the replay strategy in Table 5. Incorporating replay yields consistent performance improvements across all mathematical

*Table 5.* Ablation on the accuracy-estimation-error-prioritized replay for Qwen2.5-Math-1.5B trained on the DeepScaleR training set. We report the pass@1 performance across mathematical reasoning benchmarks, with Avg@32 for the AIME24 and AIME25 benchmarks. **Avg.** denotes the average of the performance on each of the benchmarks.

| Model | Method | MATH-500 | Olympiad | Minerva | AIME 24 Avg@32 | AIME 25 Avg@32 | Avg. |
|---|---|---|---|---|---|---|---|
| Qwen2.5-Math-1.5B | PACED-RL | 73.2 | 37.9 | 29.0 | 13.3 | 7.3 | 32.1 |
| | w/o Replay | 73.0 | 36.3 | 26.1 | 9.8 | 6.1 | 30.2 |

reasoning benchmarks, indicating that selectively revisiting question–output pairs whose predicted accuracy disagrees largely with observed accuracy provides an informative training signal for improving sample efficiency.

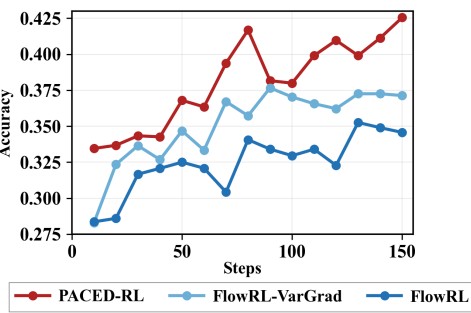

*Figure 7.* Progression of average test accuracy of FlowRL, FlowRL-VarGrad, and PACED-RL across the 5 mathematical reasoning benchmarks. Results obtained using Qwen2.5-Math-1.5B trained on the DeepScaleR dataset.

**Comparison to TB-VarGrad.** To stabilize training by circumventing the need to train the partition function, several prior works (Zhang et al., 2023; Venkatraman et al., 2024; Bartoldson et al., 2025) adopt the VarGrad variant of the Trajectory Balance objective, which replaces the learned partition function $Z_\phi(\mathbf{x})$ with a batch-level estimate. Specifically, for a prompt $\mathbf{x}_i$ and its $N$ corresponding rollouts $\{\mathbf{y}_{i,j}\}_{j=1}^N$, the batch estimate of $Z_\phi(\mathbf{x}_i)$ can be obtained as:

$$\log \hat{Z}(\mathbf{x}_i) = \frac{1}{N} \sum_{j=1}^N \left( R(\mathbf{x}_i, \mathbf{y}_{i,j}) - \log \pi_\theta(\mathbf{y}_{i,j} \mid \mathbf{x}_i) \right). \tag{10}$$

Crucially, as the estimate in Eq. 10 can only be made *after* the rollout generation phase, it cannot be used as an online accuracy estimator to guide adaptive prompt selection.

Using the Qwen2.5-Math-1.5B model, we compare PACED-RL against FlowRL-VarGrad, a variant of FlowRL that uses the batch-estimate $Z_\phi(\mathbf{x})$ for training instead of learning the partition function. Consistent with the observations of Zhang et al. (2023), Fig. 7 shows that FlowRL-VarGrad converges faster and achieves higher test performance than the original FlowRL approach. Nevertheless, PACED-RL consistently outperforms FlowRL-VarGrad throughout training,

demonstrating that explicitly learning the partition function—and leveraging the accuracy information it encodes for increased sample efficiency—is more effective than relying on a batch-estimate approximation.

## 6. Conclusion & Future Works

**Conclusion.** We introduce PACED-RL, a distribution-matching post-training framework that improves LLM reasoning performances by explicitly leveraging the learned partition function as an online accuracy signal. By exploiting the accuracy signals obtained without incurring additional compute cost, PACED-RL adaptively concentrates training resources on the most informative training samples, leading to a more effective and sample-efficient RLVR post-training for LLM reasoning. Extensive experiments on mathematical reasoning and code generation tasks across three models demonstrate the consistent effectiveness of our proposed approach. Overall, by reinterpreting the partition function as an online accuracy signal, PACED-RL offers a principled pathway towards more sample-efficient distribution-matching methods for training LLMs.

**Future Works** The interpretation of the partition function as an online accuracy estimator opens up new opportunities for inference time strategies. Beyond its role in training, future work could investigate techniques such as adaptive self-consistency, where additional samples are selectively allocated to question prompts predicted to be difficult according to the learned partition function, enabling more compute-efficient and accuracy-aware reasoning.

Additionally, we leave the study of applying GFlowNet objective variants—such as Detailed Balance (Bengio et al., 2023) and Sub-Trajectory Balance (Madan et al., 2023)—to PACED-RL in multi-step reasoning settings for future work. This extension would allow us to transition from leveraging the partition function as a question-level difficulty estimator to utilizing intermediate state *flows* as step-level difficulty signals, thereby enabling more granular credit assignment and difficulty-aware guidance in multi-step reasoning tasks.

Lastly, a promising direction for future works is to extend PACED-RL to asynchronous GFlowNet reinforcement learning frameworks (Bartoldson et al., 2025) to substantially improve scalability and training throughput.

## Acknowledgments

This work was partly supported by Institute of Information & communications Technology Planning & Evaluation (IITP) grant funded by the Korea government (MSIT) [No.RS-2022-II220184, Development and Study of AI Technologies to Inexpensively Conform to Evolving Policy on Ethics & No.RS-2021-II212068, Artificial Intelligence Innovation Hub (Artificial Intelligence Institute, Seoul National University)]. K. Jung is with ASRI, Seoul National University, Korea.

## Impact Statement

This paper presents work whose goal is to advance the field of Machine Learning. We confirm that all datasets included in our study are sourced from established, publicly available repositories and standard benchmarks. There are many potential societal consequences of our work, none which we feel must be specifically highlighted here.

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

## A. Derivations

### A.1. Proof of Proposition 4.1

For a fixed question $\mathbf{x}$, and rollouts from $\pi_{\text{old}}(\cdot \mid \mathbf{x})$, the expected loss is

$$\mathcal{L}(\mathbf{x}) := \mathbb{E}_{\mathbf{y} \sim \pi_{\text{old}}(\cdot \mid \mathbf{x})} \left[ \left( \log \frac{Z_\phi(\mathbf{x}) \, \pi_\theta(\mathbf{y} \mid \mathbf{x})}{\pi_{\text{old}}(\mathbf{y} \mid \mathbf{x}) \, \exp(\beta^{-1} r(\mathbf{x}, \mathbf{y}))} \right)^2 \right]. \tag{11}$$

By differentiating w.r.t. $\log Z_\phi(\mathbf{x})$, we have

$$\frac{\partial \mathcal{L}(\mathbf{x})}{\partial \log Z_\phi(\mathbf{x})} = \mathbb{E}_{\mathbf{y} \sim \pi_{\text{old}}(\cdot \mid \mathbf{x})} \left[ 2 \left( \log Z_\phi(\mathbf{x}) + \log \pi_\theta(\mathbf{y} \mid \mathbf{x}) - \log \pi_{\text{old}}(\mathbf{y} \mid \mathbf{x}) - \beta^{-1} r(\mathbf{x}, \mathbf{y}) \right) \right]. \tag{12}$$

Setting the derivative to zero yields

$$\mathbb{E}_{\mathbf{y} \sim \pi_{\text{old}}(\cdot \mid \mathbf{x})} \left[ \log Z^*(\mathbf{x}) + \log \pi_\theta(\mathbf{y} \mid \mathbf{x}) - \log \pi_{\text{old}}(\mathbf{y} \mid \mathbf{x}) - \beta^{-1} r(\mathbf{x}, \mathbf{y}) \right] = 0, \tag{13}$$

hence

$$\log Z^*(\mathbf{x}) = \mathbb{E}_{\mathbf{y} \sim \pi_{\text{old}}(\cdot \mid \mathbf{x})} \left[ \beta^{-1} r(\mathbf{x}, \mathbf{y}) + \log \pi_{\text{old}}(\mathbf{y} \mid \mathbf{x}) - \log \pi_\theta(\mathbf{y} \mid \mathbf{x}) \right]. \tag{14}$$

By rearranging, we have

$$\mathbb{E}_{\mathbf{y} \sim \pi_{\text{old}}(\cdot \mid \mathbf{x})}[r(\mathbf{x}, \mathbf{y})] = \beta \log Z^*(\mathbf{x}) - \beta \, \mathbb{E}_{\mathbf{y} \sim \pi_{\text{old}}(\cdot \mid \mathbf{x})}[\log \pi_{\text{old}}(\mathbf{y} \mid \mathbf{x}) - \log \pi_\theta(\mathbf{y} \mid \mathbf{x})] \tag{15}$$

$$= \beta \log Z^*(\mathbf{x}) - \beta D_{\text{KL}}\big(\pi_{\text{old}}(\cdot \mid \mathbf{x}) \,\big\|\, \pi_\theta(\cdot \mid \mathbf{x})\big). \tag{16}$$

Since reward is 0 for a wrong rollout and 1 for a correct rollout, we have

$$p_{\text{old}}(\mathbf{x}) = \mathbb{E}_{\mathbf{y} \sim \pi_{\text{old}}(\cdot \mid \mathbf{x})}[r(\mathbf{x}, \mathbf{y})]. \tag{17}$$

Substituting Eq. 17 to Eq. 16 yields

$$p_{\text{old}}(\mathbf{x}) = \beta \log Z^*(\mathbf{x}) - \beta D_{\text{KL}}\big(\pi_{\text{old}}(\cdot \mid \mathbf{x}) \,\big\|\, \pi_\theta(\cdot \mid \mathbf{x})\big). \tag{18}$$

### A.2. Generalizing to Arbitrary Binary Reward Configurations

Rather than assigning a reward value of 0 to an incorrect output and 1 to a correct output, several prior works adopt alternative reward configurations. For example, DAPO (Yu et al., 2025b) assigns a reward of -1 to incorrect outputs and a reward of 1 to correct outputs. We show that the relationship between the partition function and accuracy estimates extends naturally to arbitrary reward configurations.

Assume that a reward value of $a$ is assigned to an incorrect output and a reward value of $b$ is assigned to a correct output, where $b \neq a$. Then, we can express $\mathbb{E}_{\mathbf{y} \sim \pi_{\text{old}}(\cdot \mid \mathbf{x})}[r(\mathbf{x}, \mathbf{y})]$ as follows:

$$\mathbb{E}_{\mathbf{y} \sim \pi_{\text{old}}(\cdot \mid \mathbf{x})}[r(\mathbf{x}, \mathbf{y})] = a \, (1 - p_{\text{old}}(\mathbf{x})) + b \, p_{\text{old}}(\mathbf{x}) \tag{19}$$

$$= a + (b - a) \, p_{\text{old}}(\mathbf{x}). \tag{20}$$

Expressing in terms of $p_{\text{old}}(\mathbf{x})$ gives

$$p_{\text{old}}(\mathbf{x}) = \frac{\mathbb{E}_{\mathbf{y} \sim \pi_{\text{old}}(\cdot \mid \mathbf{x})}[r(\mathbf{x}, \mathbf{y})] - a}{b - a}. \tag{21}$$

Substituting Eq. (16) into Eq. (21) yields

$$p_{\text{old}}(\mathbf{x}) = \frac{\beta \log Z^*(\mathbf{x}) - \beta D_{\text{KL}}\big(\pi_{\text{old}}(\cdot \mid \mathbf{x}) \,\big\|\, \pi_\theta(\cdot \mid \mathbf{x})\big) - a}{b - a}. \tag{22}$$

Note that setting $a = 0$ and $b = 1$ recovers the result obtained in Eq. 18.

*Table 6.* Hyperparameters used for training on the DeepScaleR and DeepCoder datasets. Unspecified hyperparameters inherit the default configurations under the verl library.

| Hyperparameter | DeepScaleR | DeepCoder |
|---|---|---|
| Learning Rate (LR) | $1 \times 10^{-6}$ | $1 \times 10^{-6}$ |
| Gradient Clip | 1.0 | 1.0 |
| Optimizer | AdamW | AdamW |
| Weight Decay | 0.1 | 0.1 |
| Warm-up Steps | 10 | 10 |
| Global Batch Size | 128 | 64 |
| PPO Mini-batch Size | 32 | 32 |
| Micro-batch Size (per GPU) | 8 | 8 |
| Rollouts per Question ($N$) | 8 | 8 |
| Clip Ratio | 0.2 | 0.2 |
| KL Divergence Penalty | 0.0 | 0.0 |
| Entropy Coefficient | 0.0 | 0.0 |
| Rollout Temperature | 1.0 | 1.0 |
| Max Input Tokens | 1024 | 1024 |
| Max Response Tokens | 3072 | 3072 |

# B. Experimental Details

### B.1. Additional Training Details

For training on Qwen2.5-Math-1.5B and Deepseek-Distill-Qwen-1.5B models, we use a single node consisting of 2 A6000 GPUs. For training on the Qwen2.5-Math-7B model, we use a single node consisting of 4 A100 GPUs. We use the verl (Sheng et al., 2025) library for training and evalution across all runs. For all runs in the mathematical reasoning domain, we use a training budget of 150 steps. For runs on the code generation task, due to the substantial time required for training, we limit the training to 40 hours. As for the training and evaluation data, we follow the preprocessing pipeline from the official FlowRL repository[1]. Other training hyperparameters kept constant across all runs are listed in Table 6.

### B.2. PACED-RL Implementation Details

For the partition function $Z_\phi(\mathbf{x})$, following (Lee et al., 2025; Zhu et al., 2026), we use a 3-layer MLP on top of LLM last hidden layer embeddings. We use the last hidden layer embeddings of the frozen base model as the input to the $Z_\phi(\mathbf{x})$. The $Z_\phi(\mathbf{x})$ module is optimized using a seperate PyTorch optimizer, with the learning rate set to 1e-4. Lastly, after obtaining the accuracy estimates from Eq. 8, we clip the value to [0,1].

### B.3. Baseline Implementation Details

For GRPO (Shao et al., 2024) and DS (Yu et al., 2025b), we use the implementation provided in the verl (Sheng et al., 2025) library. For FlowRL (Zhu et al., 2026) and MoPPS (Qu et al., 2026), we adopt the official implementations and default hyperparameters provided by the authors in their implementations. For LILO (Foster et al., 2026), we implement the over-sampling and rejection-sampling procedure that selects the top-$m$ question prompts whose accuracies are closest to $0.5$. Following the original specification, at each training step we uniformly sample $4m$ prompts from the dataset and generate $N$ rollouts for each prompt. The $m$ prompts whose empirical accuracies are closest to $0.5$ are then selected for training, and the corresponding rollouts are reused for policy optimization.

# C. Supplementary Results

### C.1. Correlation of Accuracy Estimates to Empirically Observed Accuracies

Fig. 8 and Fig. 9 show the Spearman's correlation coefficient and Pearson's correlation coefficient for randomly sampled question prompts over the training process, on the Qwen2.5-Math-1.5B model. The constantly high ($> 0.5$) correlation values after $\sim 20$ training steps indicate that the online accuracy estimates from $Z_\phi$ serve as reliable estimators.

---

[1]https://github.com/Xuekai-Zhu/FlowRL

---

**Algorithm 1** PACED-RL

---

**Require:** dataset $\mathcal{D}$; steps $T$; batch size $m$; rollout size $N$; KL coefficient $\beta$; target accuracy $\tau$; buffer capacity $B_{\max}$; replay add count $B_{\text{add}}$
 1: Initialize policy $\pi_0$ and partition network $Z_\phi$
 2: Initialize replay buffer $\mathcal{B} \leftarrow \emptyset$ with capacity $B_{\max}$
 3: Set $\pi_{\text{old}} \leftarrow \pi_0$
 4: **for** $t = 0$ **to** $T - 1$ **do**
 5:     Estimate accuracies: $\hat{p}_{\text{old}}(\mathbf{x}) \leftarrow \beta \log Z_\phi(\mathbf{x}) \quad \forall \mathbf{x} \in \mathcal{D}$
 6:     Select $m$ questions from accuracy estimates: $\mathcal{D}_t \leftarrow \operatorname{argmin}_{\substack{\mathcal{D}_t \subseteq \mathcal{D} \\ |\mathcal{D}_t| = m}} \sum_{\mathbf{x} \in \mathcal{D}_t} |\hat{p}_{\text{old}}(\mathbf{x}) - \tau|$
 7:     Generate trajectories using $\pi_{\text{old}}$: $\mathcal{T}_t \leftarrow \{(\mathbf{x}_i, \{\mathbf{y}_{i,j}\}_{j=1}^N)\}_{i=1}^m, \quad \mathbf{y}_{i,j} \sim \pi_{\text{old}}(\cdot \mid \mathbf{x}_i)$
 8:     Augment training batch with replay: $\widetilde{\mathcal{T}}_t \leftarrow \mathcal{T}_t \cup \mathcal{B}$
 9:     Update $\pi_t$ to $\pi_{t+1}$, update $Z_\phi$ on $\widetilde{\mathcal{T}}_t$ using Eq. (6)
10:     Set $\pi_{\text{old}} \leftarrow \pi_{t+1}$
11:     Filter out incorrect trajectories from $\mathcal{T}_t$: $\mathcal{T}'_t \leftarrow \{(\mathbf{x}, \{\mathbf{y}_j\}) \in \mathcal{T}_t \mid r(\mathbf{x}, \mathbf{y}_j) = 1\}$
12:     Select $B_{\text{add}}$ samples from $\mathcal{T}'_t$ to add to buffer: $\mathcal{C}_t \leftarrow \operatorname{argmax}_{\mathcal{C}_t \subseteq \mathcal{T}'_t, |\mathcal{C}_t| = B_{\text{add}}} \sum_{(\mathbf{x}, \{\mathbf{y}_j\}) \in \mathcal{C}_t} \left| p_{\text{old}}(\mathbf{x}) - \hat{p}_{\text{old}}(\mathbf{x}) \right|$
13:     Add replay candidates to buffer: $\mathcal{B} \leftarrow \text{Push}(\mathcal{B}, \mathcal{C}_t)$
14: **end for**

---

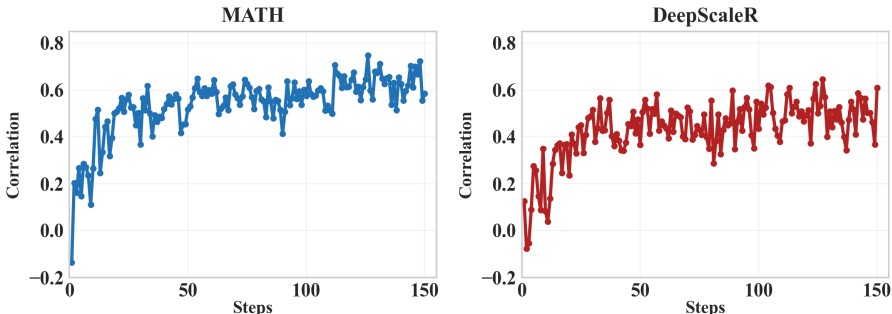

*Figure 8.* Spearman's correlation coefficient measured on the MATH training data and the DeepScaleR training data, using the Qwen2.5-Math-1.5B model.

## C.2. Robustness of the Accuracy Estimator to Adversarial Inputs

Due to the light-weighted nature of the accuracy estimator, the accuracy estimation may be sensitive to adversarial cases in which small textual perturbations induce large changes in question difficulty. To examine this failure mode, we use the MATH-Perturb benchmark (Huang et al., 2025b), which constructs two perturbed variants for each original question: *Simple*, where small edits preserve the solution structure and difficulty, and *Hard*, where small but semantically important edits change the solution and increase the difficulty. An illustrative example of a sample from the Math-Perturb benchmark is as follows:

**Original question:** Find the range of

$$y = \frac{x^2 + 3x + 2}{x + 1}.$$

**Simple variant:** Find the range of

$$y = \frac{x^2 + 3x + 2}{x + 2}.$$

**Hard variant:** Find the range of

$$y = \frac{x^2 + 3x + 2}{x}.$$

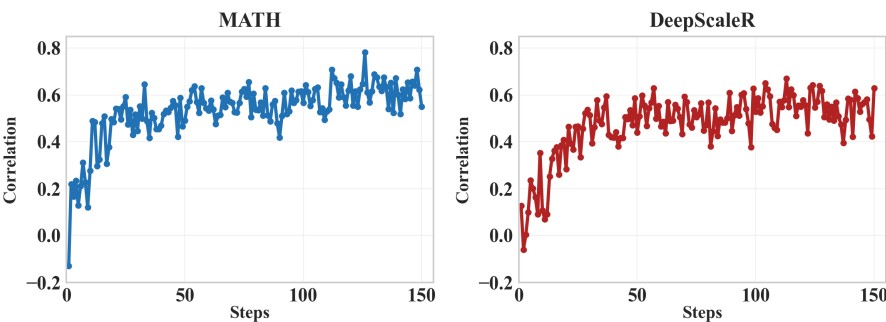

*Figure 9.* Pearson's correlation coefficient measured on the MATH training data and the DeepScaleR training data, using the Qwen2.5-Math-1.5B model..

Since the original questions are not explicitly released, we compare the *Simple* and *Hard* variants directly. Specifically, throughout training, we measure the pairwise accuracy of the estimator in Eq. 8, defined as the fraction of pairs for which the *Hard* variant is assigned a lower predicted accuracy than the corresponding *Simple* variant. As shown in Table 7, results with Qwen2.5-Math-1.5B show that the estimator correctly identifies the *Simple* variant as easier in the majority of cases throughout training, even under this adversarial perturbation setting. Taken together with the high correlations shown in Fig. 8 and Fig. 9, we view our estimator as a useful low-cost proxy for difficulty, sufficient for an adaptive learning strategy. We leave further studies on improving sensitivity to prompt perturbations for future works.

*Table 7.* Pairwise accuracy over the course of training.

| Step | Pairwise Accuracy (%) |
|------|----------------------|
| 30   | 65.2                 |
| 60   | 67.0                 |
| 90   | 67.7                 |
| 120  | 69.9                 |
| 150  | 69.5                 |

## D. Limitations

While we evaluate PACED-RL on models with 1.5B and 7B parameters and observe consistent improvements across both scales, we do not explore substantially larger models due to computational constraints. As a result, it remains an open question whether the gains of PACED-RL persist when applied to substantially larger LLMs.

In addition, although our experiments span both mathematical reasoning and code generation tasks, we focus exclusively on domains where verifiable reward signals are available. We do not study non-verifiable settings such as preference optimization or open-ended creative generation. Extending PACED-RL to these domains may require more sophisticated reward modeling or alternative accuracy estimation mechanisms, which we leave to future work.

Lastly, our low-cost accuracy estimation based on the learned partition function $Z_\phi$ relies on the assumption that the term $\beta\, D_{\mathrm{KL}}\big(\pi_{\mathrm{old}}(\cdot \mid \mathbf{x}) \,\big\|\, \pi_\theta(\cdot \mid \mathbf{x})\big)$ remains sufficiently small during training and can therefore be safely neglected. While our empirical results demonstrate that PACED-RL yields consistent performance improvements in both code generation and mathematical reasoning domains, the validity of this assumption may depend on the characteristics of the task and training dynamics. In particular, when extending PACED-RL to other domains, this approximation should be carefully examined, as deviations from this regime could degrade the accuracy of the estimator. We leave a systematic investigation of this assumption across broader domains to future work.

