# OpenReview forum: "Beyond Normalization: Rethinking the Partition Function as a Difficulty Scheduler for RLVR"
_ICML.cc/2026/Conference — ICML 2026 regular_

### Official Review · Reviewer_9vfy · 2026-03-12

**Soundness:** 2
**Presentation:** 3
**Significance:** 2
**Originality:** 3
**Overall Recommendation:** 4
**Confidence:** 3

**Summary:**

This paper introduces PACED-RL, which aims to focus RL sampling compute on the most informative prompts. The authors establish a relationship between the partition function and the expected per-question accuracy, parameterizing it within the model for estimation. This enables low-cost accuracy estimation that can guide prompt selection in reinforcement learning. Experimental results demonstrate that PACED-RL consistently outperforms previous RL and GFlowNet baselines in both accuracy and exploration efficiency, while maintaining diversity in LLM reasoning.

**Compliance With Llm Reviewing Policy:**

Affirmed.

**Final Justification:**

The author's rebuttal resolved my concerns, and I will raise my score.

**Key Questions For Authors:**

Questions:

1. Could you provide the generation response length and generation NLL loss (or entropy) metrics during RL training, and compare these between your method and the baselines?
2. Many formulas in the paper include β in the denominator; however, setting β = 0 (removing the KL term and directly maximizing reward) is applicable for GRPO and is widely used in prior research and industrial technical reports (e.g., DAPO, CISPO, GSPO). Can your theory and method be adapted to handle β = 0, or is there a natural degenerate version?
3. By predicting accuracy directly with an MLP stacked on the backbone, I suspect there could be many inaccurate predictions. For example, changing a single token in a prompt can significantly alter its difficulty. Although these cases are rare and may not affect the effectiveness of your method, I am still curious, such as the following examples, What are the accuracies of your model's predictions:

```
calculate
[
∫_1^2 1 / sqrt(x^4 + a) dx
]
where a = 0
```

vs.

```
calculate
[
∫_1^2 1 / sqrt(x^4 + a) dx
]
where a = 1
```

**Limitations:**

yes

**Strengths And Weaknesses:**

Strengths:

The paper is grounded in solid theoretical foundations, and the connection it establishes between the partition function and per-question accuracy is novel.
The paper is well-structured and clearly organized, making it easy to follow.

Weaknesses:

The paper should report metrics such as generation response length and generation NLL loss (or entropy). These indicators are crucial for understanding the training dynamics of LLM RL and could help explain the reasons behind observed performance gains.

The paper does not discuss the case of β = 0 (see question 2 below).

The focus of the paper is on improving sampling efficiency in LLM reinforcement learning. While this is useful, I think existing methods may already work well, and I feel that adding such a complex step solely to improve sampling efficiency may not be worthwhile, especially considering the need for additional hyperparameter tuning and other overhead.

---

> ### Author Rebuttal · Authors · 2026-03-30
>
> We thank the reviewer for the careful reading and the valuable suggestions, which greatly improves our paper.
>
> ---
>
> ## ***W1/Q1: On generation length and entropy***
> We agree that generation length and entropy are important metrics. Results from the Qwen2.5-Math-7B model on the DeepScaleR dataset are as follows:
>
> **1. Mean response length throughout training**
> \\begin{array}{c|cccccc}
> \\hline
> \\text{Steps} & \\text{PACED-RL} & \\text{GRPO} & \\text{FlowRL} & \\text{DS} & \\text{LILO} &
> \\text{MoPPS} \\\\
> \\hline
> 25 & 822 & 915 & 1031 & 900 & 931& 965 \\\\
> 50 & 862 & 803 & 894 & 903 &925 & 873 \\\\
> 75 & 924 & 888 & 920 & 981 &824 & 922 \\\\
> 100 & 865 & 795 & 811 & 1032 &829 & 827 \\\\
> 125 & 962 & 971 & 859 & 957 & 947& 877 \\\\
> \\hline
> \\end{array}
> Overall, it can be seen that similar to baseline methods PACED-RL leads to a stable training, with no instabilities regarding the response length.
>
> **2. Entropy dynamics throughout training**
> \\begin{array}{c|cccccc}
> \\hline
> \\text{Steps} & \\text{PACED-RL} & \\text{GRPO} & \\text{FlowRL} & \\text{DS} & \\text{LILO} & \\text{MoPPS} \\\\
> \\hline
> 25 & 0.181 & 0.176 & 0.223 & 0.179 & 0.173 & 0.188 \\\\
> 50 & 0.141 & 0.129 & 0.220 & 0.129 & 0.121 & 0.129 \\\\
> 75 & 0.123 & 0.0924 & 0.207 & 0.0697 & 0.0836 & 0.0781 \\\\
> 100 & 0.106 & 0.0606 & 0.208 & 0.0388 & 0.0406 & 0.0456 \\\\
> 125 & 0.0808 & 0.0339 & 0.211 & 0.0297 & 0.0331 & 0.0314 \\\\
> \\hline
> \\end{array}
> Overall, the entropy of FlowRL is the highest, with PACED-RL second, followed by the rest of the baselines. We interpret the lower entropy of PACED-RL relative to FlowRL as indicating a more efficient concentration on the diverse higher-quality reasoning trajectories. Crucially, this does not come at the expense of useful diversity, as PACED-RL improves on pass@k, our main diversity proxy (Figure 3).
> ## ***W2/Q2: On the case where $\beta=0$***
> **1. Effect of $\beta=0$ on the target distribution**
>
> The current optimal policy is:
>
> $\pi_{\beta}^*(y|x)=\frac{\pi_{old}(y|x)e^{r(x,y)/\beta}}{\sum_{y'}\pi_{old}(y'|x)e^{r(x,y')/\beta}}$
>
> As $\beta \to 0$, all mass will be concentrated on the correct outputs. Thus, we have:
>
> $\pi_{\beta}^*(y \mid x)\rightarrow\frac{\pi_{\mathrm{old}}(y \mid x)\mathbf{1}[r(x,y)=1]}{\sum_{y'} \pi_{\mathrm{old}}(y' \mid x)\mathbf{1}[r(x,y')=1]}$,
>
> i.e. $\pi_{\textrm{old}}$ conditioned on the correct output [1].
>
> **2. Effect of $\beta=0$ on the training objective**
>
> While $\beta=0$ induces a valid target policy, the TB objective itself will not be well-defined, since it contains the reward-induced term $\exp(r/\beta)$. At this limit, TB no longer provides a finite objective. We emphasize that the same issue also applies to the original FlowRL setting.
>
> ## ***Q3: On the effect of small prompt perturbation on accuracy estimation***
> To carry out a thorough investigation, we used the MATH-Perturb benchmark [2], which makes the following two variants of the same question:
>
> (1) Simple: Small edits that do not change the difficulty/solution to the question.
> (2) Hard: Small but fundamental edits that change the solution and increase difficulty.
>
> An illustrative example provided by authors of Math-Perturb is:
>
> Question: Find the range of $y=\frac{x^{2}+3x+2}{x+a}$.
>
> “Original” sets $a=1$, “Simple” sets $a=2$, and “Hard” sets $a=0$.
>
> Since the Original questions are not provided explicitly, we directly compared the Simple and Hard variants. Experimental results under Qwen2.5-Math-1.5B reveal that the estimator predicts the Simple variant to be easier throughout training in most cases, even under this adversarial setting.
>
> \\begin{array}{c|c}
> \\hline
> \\text{Step} & \\text{Pairwise Accuracy (\\%)} \\\\
> \\hline
> 30 & 65.2 \\\\
> 60 & 67.0 \\\\
> 90 & 67.7 \\\\
> 120 & 69.9 \\\\
> 150 & 69.5 \\\\
> \\hline
> \\end{array}
>
> Taken together with the high correlations shown in Figures 7~8, we view our estimator as a useful low-cost proxy for difficulty, sufficient for an adaptive learning strategy. We leave further studies on improving sensitivity to prompt perturbations for future works.
>
> ## ***W3: On the utility/practical value of PACED-RL***
> We would like to emphasize that the extra computational cost is negligible: the additional $Z_{\phi}$ forward passes account for only 0.0018\%--0.030\% of training-step time (Table.3). In return, PACED-RL reaches the best performance of MoPPS and DAPO in only $0.51\times$ and $0.31\times$ their wall-clock time, respectively, and improves pass@1 over GRPO/FlowRL by 29.1\%/40.0\%. It also introduces no extra tuning burden: $\tau=0.5$ is a principled default [3], and $\beta$ is a fixed constant already standard in RLVR.
>
>
> [1]: GUARANTEED GENERATION FROM LARGE LANGUAGE MODELS (ICLR 2025)
>
> [2]: MATH-Perturb: Benchmarking LLMs’ Math Reasoning Abilities against Hard Perturbations (ICML 2025)
>
> [3]: LILO: Learning to Reason at the Frontier of Learnability (Neurips 2025)
>
> ---
> We appreciate the constructive suggestions. Should you have any further questions, we are happy to assist.

---

> > ### Author Rebuttal · Reviewer_9vfy · 2026-04-02
> >
> > The author's rebuttal resolved my concerns, and I will raise my score.

---

> > > ### Author Response · Authors · 2026-04-06
> > >
> > > We are pleased that our response clarified your concerns, and we greatly appreciate the increased score. Thank you again for your thoughtful feedback and for helping strengthen the paper.

---

### Official Review · Reviewer_r9ut · 2026-03-14

**Soundness:** 3
**Presentation:** 3
**Significance:** 3
**Originality:** 3
**Overall Recommendation:** 4
**Confidence:** 3

**Summary:**

This work focuses on reinforcement learning for post-training LLMs with verifiable rewards, with a focus on GFlowNet-based distribution matching and data selection in RLVR. The paper proposes PACED-RL, which reuses the learned partition function as a per-prompt online accuracy signal and then uses that signal for adaptive prompt selection and accuracy-estimation-error prioritized replay. The empirical study covers mathematical reasoning and code generation with Qwen2.5-Math-1.5B, Qwen2.5-Math-7B, and DeepSeek-R1-Distill-Qwen-1.5B.

**Compliance With Llm Reviewing Policy:**

Affirmed.

**Final Justification:**

I believe the paper would be stronger with broader evidence on when these approximations remain valid, so I will keep my overall positive score unchanged. However, the rebuttal has increased my confidence in my assessment, and I am raising my confidence score from 2 to 3.

**Key Questions For Authors:**

Please check the weaknesses part.

**Strengths And Weaknesses:**

## Strengths

1. The core idea is simple and useful. The paper does not add a separate estimator or extra rollout phase for scheduling. Instead, it reuses a signal that is already produced during GFlowNet training. The key derivation on page 4 shows that, under the modified TB loss can be written as log of partition function minus a KL term.  And the method then uses it as a practical estimator after arguing that the KL term stays small. That is a nice reinterpretation of the partition function.

2. The empirical section is fairly broad (Sec.5). On the math benchmarks in Table 1, PACED-RL improves the Qwen2.5-Math-7B model on AIME24, while keeping the rollout budget. Figure 2 also shows wall-clock gains on code.

3. The added analysis beyond the main table is reasonable. Figure 5 on page 7 shows that the sampled prompt difficulty rises during training, which matches the intended curriculum effect.

## Weaknesses

My main concern is that the core estimator depends on dropping the KL correction term, but the paper does not fully establish when this is safe. Figure 1 shows a very small mean estimated KL term for one training setting, and the appendix later reports Pearson and Spearman correlations above 0.5 after about 20 steps for Qwen2.5-Math-1.5B. That is promising, but it is still limited evidence for a method that depends on per-prompt accuracy estimates across tasks, models, and training regimes. I would like to see stronger per-prompt calibration analysis, not only correlations.

Besides, PACED-RL replaces pi-ref with pi-old in the TB loss, which changes the training target in a meaningful way. The paper does not fully discuss whether this surrogate keeps the same desirable properties as the original FlowRL setup, or when the difference matters in practice.


(Overall, the method is clean, the main idea is easy to follow, and the experiments are solid.)

---

> ### Author Rebuttal · Authors · 2026-03-30
>
> Thank you for your thoughtful and encouraging feedback, and for the valuable suggestions to further enhance our paper.
>
> ---
>
> ## ***W1: On dropping the KL term***
> Below we provide more analyses that reveal that for reasoning tasks, dropping the KL correction term does not materially affect the sampling dynamics.
>
> **Experiment Setup**: Every 10 training steps, we randomly sample 512 prompts and compare the top-$k$ prompts whose estimated accuracies are closest to $0.5$ under two estimators: (A) the estimator without the KL term, i.e. the current implementation, and (B) the KL-corrected estimator, where the KL term is estimated by MC estimation with 8 rollouts. We report the selection overlap, $\frac{|A \cap B|}{k}$, which ranges from 0 to 1, with 1 indicating perfect agreement. As with the main experiments, we use $k=128$ and evaluate both Qwen2.5-Math-1.5B and Qwen2.5-Math-7B.
>
> **Results**:
> \\begin{array}{c|cc}
> \\hline
> \\text{Step} & \\text{Qwen-2.5-Math-1.5B} & \\text{Qwen-2.5-Math-7B} \\\\
> \\hline
> 30 & 0.984 & 0.984 \\\\
> 60 & 1.00 & 0.984 \\\\
> 90 & 0.992 & 0.992 \\\\
> 120 & 0.984 & 0.968 \\\\
> 150 & 0.976 & 1.00 \\\\
> \\hline
> \\end{array}
>
> Results show a near-perfect (>0.968, i.e. >124/128) overlap between the top-$k$ prompts selected with and without the KL correction term throughout training. Meanwhile, MC estimation of the KL term alone added overhead corresponding to 70\% of the original training-step latency for the 1.5B model and 87\% for the 7B model - even though we could consider only 512 prompts instead of the full dataset due to compute constraints. These results suggest that omitting the KL term is a practical choice for exploiting the partition function’s accuracy signal efficiently.
>
> While these results support the negligible-KL assumption for reasoning tasks, as noted in the Limitations section, its validity may depend on the task and training dynamics. We therefore caution against directly extending this assumption to other domains, such as long-horizon agentic tasks, and leave broader investigation to future work.
>
> ## ***W2: On replacing $\pi_{\textrm{ref}}$ with $\pi_{\textrm{old}}$***
> Below we provide an interpretation of replacing $\pi_{\textrm{ref}}$ with $\pi_{\textrm{old}}$, and its practical effects.
>
> In the original FlowRL formulation, the optimal policy under the following KL-regularized objective was set as the target distribution:
>
> $\max_{\theta} \mathbb{E}[r(x,y)]-\beta D_{\mathrm{KL}}(\pi_{\theta}(\cdot\mid x)||\pi_{\mathrm{ref}}(\cdot\mid x))$.
>
> To optimize the LLM policy towards the corresponding optimal policy, FlowRL used the following TB loss:
>
> $\mathcal{L}_{\mathrm{TB}}(x,y;\theta,\phi)$
>
> $=\big[\log(\frac{Z_{\phi}(x)\pi_{\theta}(y\mid x)}{\pi_{\mathrm{ref}}(y\mid x)\exp(\beta^{-1}r(x,y))})\big]^2$.
>
> As such, the substitution of $\pi_{\textrm{ref}}$ with $\pi_{\textrm{old}}$ in our TB loss can be interpreted as optimizing towards the optimal policy under the following KL-regularized objective, with the KL penalty anchored at $\pi_{\text{old}}$ instead of $\pi_{\text{ref}}$:
>
> $\max_{\theta} \mathbb{E}[r(x,y)]-\beta D_{\mathrm{KL}}(\pi_{\theta}(\cdot\mid x)||\pi_{\mathrm{old}}(\cdot\mid x))$.
>
> We note that replacing the KL anchor with the moving $\pi_{\textrm{old}}$​ may increase policy drift from $\pi_{\textrm{ref}}$. However, in practice, the difference is moderate - similar to tripling the learning rate under the original $\pi_{\textrm{ref}}$ anchor. The table below reports $D_{\mathrm{KL}}(\pi_\theta(\cdot \mid x)||p(\cdot \mid x))$ under different choices of $p$ and learning rates: $p=\pi_{\mathrm{ref}}$ with $1\times$ lr, $p=\pi_{\mathrm{old}}$ with $1\times$ lr, $p=\pi_{\mathrm{ref}}$ with $3\times$ lr, and $p=\pi_{\mathrm{ref}}$ with $4\times$ lr. Results were obtained with the Qwen2.5-Math-1.5B model.
>
> \\begin{array}{c|cccc} \\hline
> \\text{Step} & \\pi_{ref}(1\\times lr) & \\pi_{old}(1\\times lr) & \\pi_{ref}(3\\times lr) & \\pi_{ref}(4\\times lr)\\\\ \\hline
> 30 & 0.00026335 & 0.0025477 & 0.0020514 & 0.0025832 \\\\
> 60 & 0.0010988 & 0.0040306 & 0.0033553 & 0.0040061 \\\\
> 90 & 0.0015948 & 0.0043957 & 0.0049966 & 0.005661 \\\\
> 120 & 0.002068 & 0.0058649 & 0.0062784 & 0.0074294 \\\\
> 150 & 0.0023786 & 0.0070374 & 0.0073849 & 0.0092287 \\\\
> \\hline \\end{array}
>
> From a performance perspective, in FlowRL, the target optimal policy is fixed, defined w.r.t. $\pi_{\textrm{ref}}$. In contrast, by defining the optimal policy w.r.t $\pi_{\textrm{old}}$, the policy is allowed to be optimized more freely, which may contribute to the performance boosts elicited by PACED-RL compared to FlowRL. PACED-RL also maintains, or improves upon the desirable properties of FlowRL - namely output diversity - as evidenced by results shown in Figure. 3.
>
> ---
>
> We thank the reviewer for the insightful and constructive comments. We would be glad to provide further clarification if needed.

---

> > ### Author Rebuttal · Reviewer_r9ut · 2026-04-03
> >
> > Thank you for the detailed rebuttal. The additional analysis on dropping the KL term and the clarification of replacing $\pi_{\mathrm{ref}}$ with $\pi_{\mathrm{old}}$ address my main questions and make the paper’s design choices much clearer. In particular, the strong overlap in prompt selection with and without the KL correction, together with the discussion of the practical effect of using $\pi_{\mathrm{old}}$, provides useful support for the proposed method. I still believe the paper would be stronger with broader evidence on when these approximations remain valid, so I will keep my overall score unchanged. However, the rebuttal has increased my confidence in my assessment, and I am raising my confidence score from 2 to 3.

---

> > > ### Author Response · Authors · 2026-04-06
> > >
> > > We are pleased that our response helped address your concerns, and appreciate your positive assessment. As noted by the reviewer and in our Limitations section, a broader investigation into the validity of dropping the KL term across diverse domains is an important direction, which we leave for future works. We thank the reviewer again for the valuable feedback.

---

### Official Review · Reviewer_GAUV · 2026-03-18

**Soundness:** 3
**Presentation:** 2
**Significance:** 4
**Originality:** 3
**Overall Recommendation:** 5
**Confidence:** 3

**Summary:**

This paper proposes PACED-RL, a post-training reinforcement learning framework for LLMs based on GFlowNets. The key contribution is a theoretical insight: the partition function ($Z_\phi$) in GFlowNets, typically viewed merely as a mathematical normalization term, naturally encodes the real-time expected reward of a given prompt.

Leveraging this "zero-cost" accuracy estimator, the authors design two mechanisms to improve sample efficiency: (1) a difficulty-aware prompt scheduler that selects prompts with near 0.5 predicted accuracy, (2) an estimation-error-prioritized experience replay buffer. Experiments on mathematical reasoning and code generation tasks demonstrate that PACED-RL significantly reduces wall-clock training time while improving performance and maintaining generation diversity compared to strong baselines.

**Compliance With Llm Reviewing Policy:**

Affirmed.

**Final Justification:**

The authors largely address my concerns. This is a highly practical and theoretically sound paper. I am enthusiastically raising my score and strongly advocate for its acceptance.

**Key Questions For Authors:**

See Weaknesses.

**Limitations:**

No. While the authors provided a basic Limitations section (Appendix E) and an Impact Statement, they should explicitly add two critical algorithmic risks to improve scientific transparency:

1. Dataset Starvation Risk: The deterministic greedy prompt selection (Algorithm 1, Line 6) may permanently lock out valuable hard prompts if the initial accuracy estimator makes inaccurate predictions.

2. Catastrophic Forgetting Risk: Replacing the static global anchor ($\pi_{ref}$) with a moving policy ($\pi_{old}$) in the objective function (Eq. 6) removes the standard safeguard against reward hacking and general capability degradation.

**Strengths And Weaknesses:**

1. Strengths

S1. Elegant Insight that Solves a Real Bottleneck:
Repurposing the partition function into a zero-cost difficulty estimator is a brilliant idea. It effectively solves the severe computational bottleneck of existing adaptive prompt selection methods (like LILO and Dynamic Sampling), which require expensive over-sampling (generating multiple extra LLM rollouts) just to estimate prompt difficulty.

S2. Highly Convincing Wall-Clock Efficiency:
The paper successfully translates its theoretical insight into real-world engineering gains. Figure 2 and Table 3 clearly prove that PACED-RL achieves baseline-level performance in roughly half the training time. This direct reduction in wall-clock time makes the method highly practical for the community.

S3. Preservation of Generation Diversity:
Through comprehensive Pass@k evaluations (Fig. 3), the authors convincingly show that their method accelerates training without suffering from the mode collapse commonly seen in standard reward-maximizing RL, successfully maintaining the exploration benefits of GFlowNets.

2. Major Weaknesses

W1. Severe Narrative Mismatch (Overclaiming the Role of the Scheduler): The title, abstract, and introduction heavily promote the "Difficulty Scheduler" as the core driver of the performance leap. However, the ablation study in Table 2 contradicts this narrative. On the challenging AIME 24 dataset (using the 1.5B model), removing the Replay mechanism drops the score from 13.3 to 9.8—which is barely above the random-sampling FlowRL baseline of 9.5. This reveals that on hard reasoning tasks, the prompt scheduler contributes very little on its own, and the performance breakthrough is almost entirely driven by the Error-Prioritized Replay.

Actionable Suggestion: The authors must adjust the narrative to honestly reflect this reality. The Replay mechanism should be elevated to a core contribution in the Abstract and Intro. Furthermore, adding a discussion in Section 5.3 explaining why the scheduler struggles on extremely hard datasets (likely because most initial accuracies are near 0.0, making 0.5 prompts impossible to find) would turn this weakness into a deep scientific observation.

W2. Risk of "Dataset Starvation" due to Deterministic Greedy Selection: In Algorithm 1, Line 6, the prompt selection is strictly deterministic (argmin). This creates a dangerous hidden assumption: if the $Z_\phi$ estimator incorrectly predicts that a subset of highly valuable, hard prompts has an accuracy of 0.0 at the beginning of training, those prompts will never be selected. Consequently, the model will never generate rollouts for them, and they will never enter the Replay buffer to correct the estimator's blind spots. The model could permanently lock itself out of valuable data.

Actionable Suggestion: To prevent this deadlock, the authors should introduce a soft-sampling exploration mechanism (e.g., $\epsilon$-greedy or temperature-scaled softmax) in Line 6. Additionally, please report the "Dataset Coverage Rate" (the percentage of unique training prompts actually sampled at least once across the 150 steps) in the appendix to prove the model is not collapsing onto a small subset of the data.

W3. Unverified Risk of Catastrophic Forgetting (The Moving KL Anchor): To make the mathematical derivation work (Equation 6), the authors replace the static base model ($\pi_{ref}$) with the continuously updating policy ($\pi_{old}$) in the KL divergence penalty. In standard RLHF/RLVR, the static $\pi_{ref}$ acts as a crucial anchor to prevent the model from losing its general language capabilities (reward hacking). By turning this into a sliding window, the safety anchor is removed.

Actionable Suggestion: The authors need to provide a brief evaluation on a general capability benchmark (e.g., MMLU) in the appendix to prove that this modification does not lead to catastrophic forgetting during the training process.

3. Minor Weaknesses (Clarity and Presentation)

M1. Mathematical Contradiction Between Equation 9 and Algorithm 1: In Section 4.4, Equation 9 calculates the priority as N_correct / N - p_old(x) (without an absolute value), meaning it only prioritizes "false negatives" (where the model succeeded but the estimator predicted failure). However, Algorithm 1 Line 12 clearly uses an absolute value: |p_old(x) - \hat{p}_{old}(x)|.

M2. Confusing and Overloaded Notation for Accuracy: The notation $p_{old}(x)$ is heavily overloaded. In Section 4.1 (Eq. 7), it represents the theoretical expected reward. But in Section 4.4 and Algorithm 1, the text mixes up the theoretical accuracy, the predicted accuracy ($\hat{p}_{old}$), and the empirical accuracy observed from the current batch's rollouts.

M3. Abrupt Transition to Equation 6: The jump to Equation 6 (Line 166: "Specifically, we replace $\pi_{ref}$ with $\pi_{old}$") is too sudden and lacks physical motivation.

---

> ### Author Rebuttal · Authors · 2026-03-30
>
> Thank you for your positive feedback about the significance of our work and for the valuable suggestions that improve the comprehensiveness of our results.
>
> ---
>
>
> ## ***W1: On the role of the scheduler***
> Thank you for the helpful observation. In PACED-RL, both the difficulty scheduler and the accuracy-error replay are designed to exploit the partition function’s accuracy signal, and our results show that both contribute meaningfully to the overall performance gains. We have revised the Abstract and Introduction to present replay more clearly as a core component of the method.
>
> Regarding the concern that hard prompts may be excluded from training, please kindly refer to Figure 5. As training progresses, the policy is exposed to increasingly difficult prompts, and in later stages it trains on the hardest subsets of the dataset.
>
> ## ***W2/L1: On "Dataset Starvation"***
> Below we present the experimental results regarding 1) Soft-Sampling and 2) Dataset Coverage Rate.
>
> **1. On Soft-Sampling:**
>
> **Experiment Setup:** We evaluate a variant of PACED-RL that replaces the top-$k$ prompt selection with soft sampling. Prompts are sampled from a softmax distribution with logits $p(1−p)$, where $p$ is the estimated accuracy, thus assigning higher sampling probability to prompts whose estimated accuracies are closest to 0.5. We utilized the Qwen2.5-Math-1.5B model and the DeepScaleR dataset.
>
> **Results:**
> \\begin{array}{c|cccc}
> \\hline
> & \\text{greedy(current implementation)} & \\text{softmax (temp=0.4)} & \\text{softmax (temp=0.7)} & \\text{softmax (temp=1.0)} \\\\
> \\hline
> \\text{MATH500 Avg@8} & \\mathbf{73.2} & 71.8 & 71.4 & 72.2 \\\\
> \\hline
> \\end{array}
> We can observe that using soft-sampling degrades performance. We interpret the cause of this phenomenon as soft-sampling leading to sub-optimal prompts being used for training, leading to sub-optimal results.
>
> We also note that as in our response to [W1], the LLM policy is trained with increasingly more difficult prompts. At the latter stages of training, the LLM policy trains on the most difficult subsets of the dataset.
>
> **2. On Dataset Coverage Rate:**
>
> **Experiment Setup:** We measure coverage—the percentage of distinct questions encountered during training relative to the no-adaptive-sampling variant — at step 150, as well as the total number of distinct prompts seen by step 150. We utilized the Qwen2.5-Math-1.5B, 7B model and the DeepScaleR dataset.
>
> **Results:**
> \\begin{array}{c|cc}
> \\hline
> & \\text{Qwen2.5-Math-1.5B} & \\text{Qwen2.5-Math-7B} \\\\
> \\hline
> \\text{Coverage (\\%)} & 55.7 & 62.3 \\\\
> \\text{Total Distinct Prompts} & 10695 & 11962 \\\\
> \\hline
> \\end{array}
>
> As shown, PACED-RL trains on a subset of prompts seen by the no-adaptive-sampling variant. It selectively filters out less useful prompts to avoid wasted trajectory generation and optimization. Moreover, the total number of distinct prompts remains large (>10k), indicating that PACED-RL does not collapse to a narrow subset of the dataset.
>
>
> ## ***W3/L2: On catastrophic forgetting***
> Below, we provide results on general capability benchmarks and on the effect of modifying the KL anchor.
>
> **Experiment Setup:** Due to compute constraints, we evaluate Qwen2.5-Math-1.5B (trained with DeepScaleR) on a subset of MMLU consisting of 5,000 randomly sampled non-math questions throughout training.
>
> **Results:**
> \\begin{array}{c|c}
> \\hline
> \\text{Step} & \\text{Avg@8} \\\\
> \\hline
> 0   & 36.54 \\\\
> 30  & 36.67 \\\\
> 60  & 37.00 \\\\
> 90  & 39.26 \\\\
> 120 & 38.23 \\\\
> 150 & 40.30 \\\\
> \\hline
> \\end{array}
>
> Results indicate that general, non-math capabilities are not lost. The performance on MMLU instead steadily increases, alluding to the potential of PACED-RL generalizing to out-of-distribution tasks.
>
> We do note that replacing the KL anchor with $\pi_{\textrm{old}}$​ may increase policy drift from $\pi_{\textrm{ref}}$. However, in practice, its empirical effect is moderate - similar to tripling the learning rate under the original $\pi_{\textrm{ref}}$ anchor. For the experimental details and results, please kindly refer to our response to reviewer r9ut's concern on **"W2: On replacing $\pi_{\textrm{ref}}$ with $\pi_{\textrm{old}}$"**.
>
> ## ***M1/M2/M3: On clarity and presentation***
> Thank you for raising these issues. We will revise the paper in the final version to enhance readability so as to reduce confusion for the readers. As for **[M3]** - your concern on the physical motivation and the interpretation on replacing $\pi_{\textrm{ref}}$ with $\pi_{\textrm{old}}$ - please kindly refer to our response to reviewer r9ut's concern on **"W2: On replacing $\pi_{\textrm{ref}}$ with $\pi_{\textrm{old}}$"**.
>
> ---
>
> We greatly appreciate your positive feedback, and the helpful comments. If you have any further questions, we would be happy to help.

---

> > ### Author Rebuttal · Reviewer_GAUV · 2026-04-02
> >
> > I would like to highly commend the authors for an exceptionally rigorous, transparent, and data-driven rebuttal. The authors did not shy away from the hard questions and provided empirical evidence that directly dismantled my core concerns.
> >
> > Catastrophic Forgetting Resolved: Providing the MMLU trajectory was exactly the 'silver bullet' needed. The fact that the MMLU score steadily improved (36.54 $\rightarrow$ 40.30) puts the 'reward hacking' concern to rest within this training horizon. Furthermore, mapping the policy drift of the moving $\pi_{old}$ anchor to an equivalent '3x learning rate multiplier' under the static anchor is a brilliant piece of quantitative intuition.
> >
> > Dataset Starvation Addressed: The newly provided Dataset Coverage Rates (~56-62%, translating to >10,000 distinct prompts) are excellent. They completely alleviate my fear that the deterministic greedy scheduler might lock the model into a narrow information silo. Additionally, the ablation showing that soft-sampling actually degrades performance elegantly justifies the authors' original greedy design choice as a practical optimum.
> >
> > 1) Narrative & Clarity: I highly appreciate the authors' intellectual honesty and commitment to elevating the 'Error-Prioritized Replay' mechanism in the abstract and introduction, as well as fixing the notation inconsistencies.
> >
> > 2) KL Approximation: The prompt overlap matrix showing >96.8% agreement completely justifies dropping the exact KL term, proving it saves ~80% computational overhead with near-zero degradation in prompt selection quality.
> >
> > Given that the authors have successfully transformed all the 'theoretical risks' I pointed out into 'empirically proven robust features', the core contribution of this paper is shines through, which repurposing the partition function into a zero-cost difficulty estimator to slash training time.
> >
> > This is a highly practical and theoretically sound paper. I am enthusiastically raising my score and strongly advocate for its acceptance.

---

> > > ### Author Response · Authors · 2026-04-06
> > >
> > > We are glad that our response helped address your concerns. The increased score is very encouraging. Thank you again for your thoughtful feedback and for helping us strengthen the paper.

---

### Decision · Program_Chairs · 2026-04-30

**Decision:**

Accept (regular)

**Comment:**

This paper proposes PACED-RL, a GFlowNet-based RL post-training method that reuses the partition function as an online per-prompt accuracy signal, enabling adaptive prompt selection and prioritized replay at very low additional cost. Reviewers agreed on the main strengths of the paper: the core idea is simple and appealing, the method is practically useful, and the experiments show consistent gains in both sample efficiency and wall-clock efficiency while preserving diversity.

The main concerns in the initial reviews were about the estimator approximation, the use of a moving KL anchor, and the possibility that greedy scheduling could reduce prompt coverage. After reading the rebuttal and the subsequent reviewer updates, I believe these concerns have been sufficiently addressed within the scope of the paper. The additional analyses on prompt-selection overlap with and without the KL correction, the discussion of the moving-anchor objective, and the new coverage/general-capability results substantially strengthen the paper. Importantly, the reviewers who raised these points explicitly indicated that their concerns were resolved.

In my view, the remaining limitations are mainly about breadth of validation across settings rather than a lack of technical soundness for the setting considered here. Given the clean core insight, the practical value of the method, and the positive post-rebuttal consensus, I recommend accepting this paper.